# Meningococcal Disease in Pediatric Age: A Focus on Epidemiology and Prevention

**DOI:** 10.3390/ijerph19074035

**Published:** 2022-03-29

**Authors:** Giada Maria Di Pietro, Giulia Biffi, Massimo Luca Castellazzi, Claudia Tagliabue, Raffaella Pinzani, Samantha Bosis, Paola Giovanna Marchisio

**Affiliations:** 1Pediatric Highly Intensive Care Unit, IRCCS Ca’ Granda Foundation, Policlinico Hospital, 20122 Milan, Italy; giada.dipietro@policlinico.mi.it (G.M.D.P.); claudia.tagliabue@policlinico.mi.it (C.T.); raffaella.pinzani@policlinico.mi.it (R.P.); samantha.bosis@policlinico.mi.it (S.B.); 2Department of Pathophysiology and Transplantation, University of Milan, 20122 Milan, Italy; giulia.biffi@unimi.it; 3Pediatric Emergency Department IRCCS Ca’ Granda Foundation, Policlinico Hospital, 20122 Milan, Italy; luca.castellazzi@policlinico.mi.it

**Keywords:** invasive meningococcal disease, children, adolescents, meningococcal vaccines, immunization program

## Abstract

Meningococcal disease is caused by *Neisseria meningitidis*; 13 serogroups have been identified and differentiated from each other through their capsular polysaccharide. Serotypes A, B, C, W, X, and Y are responsible for nearly all infections worldwide. The most common clinical manifestations are meningitis and invasive meningococcal disease, both characterized by high mortality and long-term sequelae. The infection rate is higher in children younger than 1 year and in adolescents, who are frequently asymptomatic carriers. Vaccination is the most effective method of preventing infection and transmission. Currently, both monovalent meningococcal vaccines (against A, B, and C serotypes) and quadrivalent meningococcal vaccines (against serogroups ACYW) are available and recommended according to local epidemiology. The purpose of this article is to describe the meningococcal vaccines and to identify instruments that are useful for reducing transmission and implementing the vaccination coverage. This aim could be reached by switching from the monovalent to the quadrivalent vaccine in the first year of life, increasing vaccine promotion against ACYW serotypes among adolescents, and extending the free offer of the anti-meningococcal B vaccine to teens, co-administering it with others proposed in the same age group. Greater awareness of the severity of the disease and increased health education through web and social networks could represent the best strategies for promoting adhesion and active participation in the vaccination campaign. Finally, the development of a licensed universal meningococcal vaccine should be another important objective.

## 1. Introduction

Neisseria meningitidis (NM), a Gram-negative diplococcus, is the bacterium responsible for the meningococcal disease, one of the main sources of community-acquired sepsis and meningitis in the pediatric age [1]. Although quite rare, meningococcal disease constitutes a global health problem in children, with an overall mortality of around 8% and possible neurological sequelae [2].

This article focuses on the epidemiology of meningococcal disease and its prevention through vaccination in children and adolescents, focalizing more on developed countries (Europe and North America) and less on low-income countries, especially Africa. 

## 2. Epidemiology

Based on the polysaccharide capsule, 13 capsular groups of NM have been identified. Of these, only six (A, B, C, W, X, and Y) are most frequently associated with diseases in humans [3]. 

Humans are the only reservoir for the bacterium, which resides primarily in the nasopharynx of asymptomatic subjects and can be transmitted to close contacts via respiratory droplets [4]. 

Carriage is highest in teens and young adults, especially as their lifestyle involving greater close contact, whereas carriage rates are lower in infants and older adults [5]. Nasopharyngeal colonization involves approximately 10% of adults, with an increase to 24% during adolescence [4].

Meningococcal disease occurs when the pathogen enters the bloodstream, leading to systemic infection, including meningitis, sepsis (meningococcemia), or both [6].

Generally, meningococcal disease is characterized by rapid evolution and severity and requires prompt recognition and treatment. However, even when adequate treatment is started, it is associated with a negative prognosis, as between 10% and 15% of subjects die, and up to 60% have long-term sequelae [7,8,9,10,11].

The annual incidence of meningococcal disease in the United States (US) ranges between 0.11 and 1.5 cases per 100,000 persons [12]. In Europe, the incidence is 1.01 cases per 100,000 population [13]. Currently, there is wide geographical variation in the distribution of the different capsular groups of NM causing invasive disease. 

Serogroup W organisms are predominant throughout most of Africa, accounting for 44–98% of cases [14]. In the known “meningitis belt”, a region between Ethiopia and Senegal, serogroup A is more prevalent [15]. 

Furthermore, serogroups A and C are responsible for large epidemics in Africa and Asia, whereas serogroups B and C especially cause disease in Europe and the Americas [16]. Particularly, in Europe, even though there is an increasing trend in serogroups W and Y, serogroup B continues to be the main cause of invasive meningococcal disease [17]. In North America, serogroups B, C, and Y predominate [3]. In Japan, serogroup Y remains an important cause of meningococcal disease [15].

Interestingly, the incidence of meningococcal disease varies according to the age group considered, with a double peak distribution, on the one hand in the first years of life and on the other hand among adolescents and young adults [12,18]. In the US, meningococcal disease in infants <1 year of age is due to serogroup B in about 60% of cases, whereas serogroup C, Y, and W account for about 66% of cases in children older than 11 years [3]. In a recent Italian survey between 2011 and 2017, the incidence of invasive meningococcal disease increased from 0.25 cases to 0.33 cases/100,000 in 2017. Serogroup B had the highest prevalence in children less than 5 years of age. Furthermore, serogroups W and Y cases increased over the study period [19].

## 3. Vaccines

For the last 40 years, vaccines, including the polysaccharide capsule of single or multiple meningococcal serogroups, have been available. These vaccines elicit a B cell response, with the production of specific antibodies, but do not stimulate long-term memory even after repeated doses, which create an antibody concentration that is lower than those induced after primary immunization. Due to their ineffectiveness related toshort-term protection, the inability to induce immune memory, and the poor response to booster doses, polysaccharide-conjugate vaccines were developed through the conjugation of the meningococcal polysaccharide antigen to protein carriers. These vaccines invoke a T cell response that is able to induce immunological memory; stimulate a higher antibody concentration with a longer duration of protection, especially after the booster doses; and prevent nasopharyngeal carriage, inducing herd protection. Conjugate vaccines may enhance immunity against the pathogen from which the carrier protein is derived (tetanus toxoid, TT; diphtheria toxoid, DT; cross-reacting material 197, CRM, a non-toxic mutant of diphtheria toxin) [5,20].

In developed countries, where the incidence of meningococcal disease is low, vaccine efficacy is evaluated through the demonstration of specific immune responses, whereas in low-income countries, it is measured through a real reduction in disease incidence. Serum bactericidal antibody assays with human complement (hSBA) or with baby rabbit complement (rSBA) are used to measure functional antibodies against serogroups of meningococcal antigens. Immunogenicity is assessed as the proportion of persons who achieve an SBA titer above a predefined threshold or fourfold rise in SBA titers for the tested serogroups [21,22].

### 3.1. The Monovalent Meningococcal Conjugate Vaccines

The meningococcal C (MenC) conjugate vaccine was developed in 1999. Actually, there are 2 MenC-CRM (Meningitec, Nuron Biotech Inc., Exton, PA, USA and Menjugate, GlaxoSmithKline Biologicals SA, Rixensart Belgium), both conjugated to diphtheria protein cross-reactive material 197 and 1 MenC-TT (NeisVac-C, Pfizer Inc., New York, NY, USA) conjugated to tetanus toxoid. 

The United Kingdom (UK) was the first country to license the vaccine for meningococcus serogroup C in the vaccination schedule, achieving a decrease of 86.7% in the incidence of serogroup C infection in the targeted age groups from 1999 to 2001 and also a reduction of 34–61%, depending on age, in the disease incidence among unvaccinated children. The MenC conjugate vaccines are not only immunogenic but also safe. At the time of approval, in the UK, it was given from 2 months of age in 3 doses spaced one month apart in the first year of life. Many studies demonstrated that the effectiveness declined by 81% within a year. Another dose at 12 months of age was found to induce immunological memory and decrease the number of carriers, with a consequent reduction in the incidence of the disease among the unvaccinated (herd immunity). A single dose was recommended for catch-up vaccination in children 1 year and older who were not previously vaccinated [5,23,24,25].

From July 2016, as the levels of MenC disease were so low that children younger than 12 months of age were well protected by herd immunity, infants have received a single dose of a combination conjugate vaccine that includes Haemophilus influenzae type B (Hib/MenC vaccine Menitorix) at 12 months of age [26].

In the US, the MenC vaccine is not administered due to the high levels of invasive meningococcal disease (IMD) caused by other serogroups contained in the MenACWY vaccine. In Italy, a single dose of the MenC vaccine between 13 and 15 months of life is approved (with the possibility of children at increased risk of the meningococcal disease receiving 3 doses at 3, 5, and 11 months of age); it is co-administered with the measles, mumps, rubella, and varicella vaccine (MMRV) [27].

The World Health Organization (WHO) recommends the monovalent C meningococcal vaccine for all toddlers at one year of age as part of routine immunization and for subjects who have had the meningococcal disease. Infants aged 2–11 months receive 2 doses at a 2-month interval, followed by a third dose about 1 year later while children older than 12 months of age receive a single dose [28].

The introduction of the monovalent C meningococcal vaccine in immunization programs across Europe in 1999 resulted in a rapid reduction in the proportion of MenC disease, followed by a stable rate from 2013 to 2017. Among the combined 29 European Union countries reporting surveillance data, the rate of IMD cases declined from 1.9 to 1.1 cases per 100,000 population between 1999 and 2007 [29].

In 2017, of the 2979 IMDs with an identified serogroup that were reported to the European Centre for Disease Prevention and Control (ECDC), 16% were due to the C strain, scattered especially across central and southern Europe. Starting from 2011, serogroup B remains the main cause of IMD among most European countries [18].

In the UK, the largest rate of decline was observed in studies conducted after 2006, after the introduction of the meningococcal vaccine, with a drop of 78–87% in MenC cases in toddlers <1 year of age, 70–98% in subjects 1–4 years of age, and 79–93% in subjects <18/20 years of age [26].

In Italy, since 2013, vaccination coverage with the MenC vaccine within 24 months of life has risen from 77% to about 85%. In 2019, 189 cases of IMD were reported; in 2018 and 2017, 170 and 197 cases were registered, respectively. In 2019, the incidence of IMD was higher in infants <1 year (notification rate of 2.97 per 100,000) and in children aged 1–4 years (notification rate 0.88 per 100,000). For age groups in the range from 0 to 4 years, MenC was responsible for 15% of all cases, with MenB infections prevailing over the other serogroups. Whereas among young adults aged 15–24 years, the notification rate remained stable (0.58 per 100,000) from 2017 to 2019, with 31% of the confirmed cases due to serogroup C [30].

Maiden et al. demonstrated a reduction in serogroup C oropharyngeal carriage in the 2 years following the introduction of the MenC vaccine in the UK, which contributed to the achievement of herd immunity in the population, without serogroup replacement [31,32].

The MenC vaccines can be administered at the same time (using a different injection site) as any of the following vaccines: Inactivated polio vaccine (IPV); oral polio vaccine (OPV); diphtheria and tetanus vaccine alone (D or T), in combination (DT or dT), or in combination with whole cell or acellular Pertussis vaccine (DTwP or DTaP); hepatitis B vaccine (HBV); Hemophilus influenzae type B conjugate vaccine (Hib alone or in combination with other antigens); 7-valent pneumococcal conjugate (PCV7); 10-valent pneumococcal conjugate (PCV10); 13-valent pneumococcal conjugate (PCV13); and the combined measles, mumps, and rubella vaccine (MMR). In Italy, the MenC vaccine is administered concomitantly with the measles, mumps, rubella, and varicella vaccine (MMRV) [33,34]. 

The meningococcal serogroup A polysaccharide-tetanus toxoid conjugate vaccine (MenA-TT, MenAfriVac Serum Institute of India Ltd., Pune, India) was expressly designed to control the spread of the MenA disease in the African meningitis belt. It was introduced in 2010 with 2 formulations: the first for children aged between 3 and 24 months and the second for people aged 1 to 29 years. From 2010 to 2015, the incidence of suspected meningitis cases declined by 57% in vaccinated populations, with a corresponding reduction of 59% in epidemics. In fully vaccinated people, the incidence of confirmed group A disease was reduced by more than 99%. Vaccination with MenA-TT has also been reported to decrease the carriage of MenA [5,35].

### 3.2. The Quadrivalent Meningococcal Conjugate Vaccines

Quadrivalent (A, C, Y, and W135) vaccines have been approved since 2005 for use in children and adults in several countries in the world. Currently, four formulations of quadrivalent meningococcal conjugate vaccines are available worldwide, as shown in Table 1 [36,37,38,39]. Many studies have demonstrated their immunogenicity and safety [5,21,40].

In the US, the Advisory Committee on Immunization Practices (ACIP) recommends a single dose of MenACWY for all subjects aged 11–12 years and, since 2010, a booster dose for adolescents aged 16 years. Children who received MenACWY at 10 years of age do not need a second dose at 11–12 years but should obtain the booster dose at 16 years of age. Children vaccinated with MenACWY before 10 years of age and with no identified risk factors for the development of meningococcal disease should receive the first dose at 11–12 years and a second dose at 16 years of age. Adolescents who were vaccinated at 13–15 years should receive a booster dose at 16–18 years with an interval of at least 8 weeks between the doses. For teens who received the first dose after 16 years, a booster dose is not necessary, unless they are at increased risk of developing meningococcal disease. ACIP recommends the MenACWY vaccine for subjects older than 2 months of age at increased risk of meningococcal disease caused by serogroups A, C, W, or Y [21]. For more details, see Table 2.

In the US, the MenACWY program was associated with a reduction in the incidence of MenC and MenY diseases among teens and young adults [41].

In the US, the effectiveness of a single dose of MenACWY-D (Menactra) among adolescents is estimated at 69% in the 8 years after vaccination; this percentage decreases from 79% in the first year to 61% in the 8th year postvaccination. Since the introduction of MenACWY in 2017, a decrease of more than 90% in meningococcal disease incidence due to serogroups C, W, or Y, was observed among adolescents. After the primary and booster MenACWY dose were introduced, the incidence rate of meningococcal disease due to serogroup C, W, or Y declined by nearly 2-fold to 3-fold in vaccinated adolescent age groups in the US. In US countries where the MenACWY vaccination coverage is high, the carriage prevalence of meningococcal serogroups C, W, or Y combined among college students is extremely low (<1%) [21].

Mbaeyi et al. demonstrated that even if infants <1 year of age have experienced the greatest absolute decline in disease incidence due to serogroups C, W, and Y, adolescents aged 11–15 years have shown the greatest percentage decrease in disease incidence due to serogroups C, W, and Y during the post-vaccination period (a 67.0% reduction in the disease incidence during the post-primary dose period; 88.8% during the post-booster dose period) [42].

In the UK, the MenACWY is routinely offered to children aged 13–15 years. 

Ten years after the introduction of the MenC conjugate vaccine, Europe experienced many outbreaks of IMD due to meningococcal strains not contained in the vaccine. This led to the emergency implementation of a MenACWY conjugate vaccine program from 2015. In 2017, the European Surveillance System reported a rate of 0.6 confirmed cases per 100,000 population of IMD in 30 European member states, of which 96% were confirmed cases caused by serogroups B, C, W, and Y (serogroup B accountable for 51%); however, the contribution of each serogroup varied across the individual European countries. The notification rate was 8.2 per 100,000 population in infants under 1 year of age, 2.5 in children aged 1–4 years, and 1.0 in adolescents aged 15–24 years. Moreover, since 2013, a gradual decline has been observed in all children aged under 15 years while a stable rate has been described in adults. Recently, in many European countries, an increase in cases of IMD due to serogroups W and Y has been observed, with incidence rates of 17% and 12%, respectively, reported in 2016, and the highest fatality rate reported for serogroup W [18,25].

Among university students, Read et al. demonstrated that 2 months after immunization, those vaccinated against MenACWY had a lower carriage prevalence than the controls for both serogroup Y (39% carriage reduction) and serogroups C, W, and Y combined (36% carriage reduction). The authors concluded that the MenACWY vaccine reduced meningococcal carriage rates during the 12 months after vaccination, with a positive effect on transmission in the case of the implementation of the vaccination program [43]. 

In Italy, since 2017, vaccination with a quadrivalent meningococcal conjugate vaccine (MenACWY) has been recommended with a single dose at 12–14 years, both for unvaccinated subjects and for children who had already been immunized at infancy with MenC or MenACWY. In children at increased risk, the quadrivalent vaccine can also be administered in place of the MenC conjugate vaccine at 13–15 months, or as a single dose from the second year of life in those who have never received MenC conjugate vaccine can be used [27].

In Italy, in 2019, the vaccination coverage among adolescents was about 52.4% for the MenC vaccine and approximately 75% for the MenACWY vaccine. The number of IMD cases due to C, W, and Y decreased in 2018 and 2019, but, at the same time, from 2016, an increase in cases related to serogroup W has been observed, which is in line with the trends reported in other European countries in recent years. The first isolation of serogroup X was reported in 2009 [30,44].

Currently, based on local epidemiology, many countries recommend MenC instead of the MenACYW vaccine in the first years of life (Europe or Canada). In Europe, after its introduction into the national routine childhood immunization program for vaccination against serogroup C, there was a reduction in the absolute number of cases of IMD but an increase in other serogroups not contained in the monovalent vaccine. In recent times, due to the increase in the number of cases related to serogroup W, a switch from the monovalent to the quadrivalent vaccine was observed in many vaccination schedules. Thus, among those countries that have endorsed the meningococcal vaccine in pediatric age, some recommend a mixture of monovalent and quadrivalent vaccines according to the age group (e.g., Italy, Spain, and the UK) while others exclusively offer the monovalent C vaccine (e.g., Belgium, France, and Germany) or quadrivalent vaccine (e.g., the Netherlands) [45].

### 3.3. Vaccines against Serogroup B

Currently, two vaccines are available for preventing infection disease caused by *Neisseria meningitidis* serogroup B (MenB). Both vaccines are based on recombinant proteins identified by reverse vaccinology (BexseroTM) and proteomics (TrumenbaTM) approaches, respectively. Bexsero (4CMenB; Novartis Vaccines and Diagnostics, Siena, Italy) is a quadrivalent recombinant vaccine developed through “reverse vaccinology”, a new technique that allows the preparation of a vaccine starting from the identification of genes that encode antigenic proteins. This technique consists of two fundamental moments: the characterization of the *N. meningitidis* genome and the identification of common proteins present in all serotypes of group B. The new protein-based vaccine against MenB is a multicomponent vaccine that contains four immunogenic proteins: three purified recombinant antigenic proteins of *N. meningitidis* serogroup B (factor H-binding protein, fHbp; Neisserial heparin-binding antigen, NHBA; and Neisseria adhesin A, NadA) and one protein derived from vesicles of the outer membrane (OMV) of the bacterium. Trumenba (MenB-FHbp; Pfizer Inc., Philadelphia, PA, USA), on the other hand, is a bivalent recombinant vaccine that contains two variants (A and B) of the complement factor H-binding protein (fHbp).

The path to the development of an effective MenB vaccine was very complicated. The earliest attempts to develop MenB vaccines were made from 1900 to 1940 in response to infectious outbreaks during both World Wars, but excess reactogenicity due to the presence of large amounts of lipooligosaccharide (LOS) resulted in their failure. Subsequent evolution resulted in the formulation of capsular polysaccharide vaccines, which, however, were not only unsuccessful but were also shown to have a potential risk regarding the formation of autoantibodies that can cross-react with brain tissue. Indeed, the polysaccharide capsule of *N. meningitidis* B contains a polysialic acid whose antigenic structure resembles the cell surface glycoproteins of human neurological tissue. Furthermore, the capsular polysaccharide has very low immunogenicity, which cannot be modified by the conjugation process with a transport protein [46,47,48,49].

According to the European Medicines Agency (EMA) and the Food and Drug Administration (FDA), Bexsero is now indicated for active immunization starting from 2 months of life while Trumenba is licensed in those aged ≥10 years. Both vaccines are safe and well-tolerated, with usually mild or moderate adverse reactions. The most frequently reported reactions are pain, redness, and swelling at the injection site; headache; malaise; fatigue; chills; diarrhea; muscle and joint pain; and nausea [49].

According to the full product information published by the pharmaceutical company, both Bexsero and Trumenba can be administered simultaneously with routinary vaccines. In particular, Bexsero can be co-administered with hexavalent vaccine (DTaP-HBV-IPV/Hib), PCV7, MMRV, and meningococcal serogroups A, C, W, and Y vaccines. Conversely, Trumenba can be co-administered with DTaP, HPV4, and meningococcal serogroups A, C, W, and Y. However, to date, in most vaccination schedules, Bexsero is not routinely co-administered with other vaccinations [50,51,52].

According to the Center for Disease Control (CDC), Bexsero and Trumenba are not interchangeable. However, if necessary, CDC recommends waiting at least one month after the administration of one brand before administering the other one, of which the full series is needed [53].

In Europe, the highest incidence of IMD caused by MenB is in the first 5 years of life (70% of all cases). Particularly, most cases are reported in the first year of age (over 20% of all cases, with 30% of deaths), with the highest incidence reported between 4 and 8 months of life. The notification rate is 8.2 confirmed cases per 100,000 population in infants <1 year of age and 2.5 confirmed cases per 100,000 population in 1–4-year-olds. The second peak of incidence of IMD is among adolescents (15–24-year-olds), with a rate of 1.0 per 100,000. Based on IMD’s epidemiology, to obtain the highest effectiveness, the MenB vaccine should be recommended for children (in particular infants) and adolescents as the preferred target population. Furthermore, regarding immunization in the pediatric age, to establish optimum coverage and long-term protection (good immune memory), many studies have demonstrated that MenB vaccines should be offered to all infants in the first months of life, with a booster dose provided in the second year of life [54,55,56].

Despite these data, globally, MenB vaccination strategies differ widely. For example, in Australia, the MenB vaccine is recommended during routine childhood immunization (infants 1–4-year-olds) and for those aged 15–20 years. In the US, it is not currently recommended as part of children and mass immunization programs, and it is not approved in children <10 years old. Since 2015, ACIP has routinely recommended the use of meningococcal group B vaccines for people at high risk of IMD and for adolescents aged between 16 and 23 years who are not at increased risk. Despite these recommendations, in the US, this vaccination is not routinely used in this age group because of the current historically low meningococcal disease burden. In this country, this vaccine has only been used reactively to fight sporadic outbreaks, implementing a targeted mass immunization program during these “emergence situations” to interrupt person-to-person transmission. In these cases, CDC recommends those at increased due to an outbreak who have previously received the full vaccine series also receive a booster shot risk [25,53,57].

Despite these recommendations, some experts suggest the vaccination of high-risk children with Bexsero over Trumenba due to its lower reactogenicity. In Canada, similar to the ACIP recommendations, vaccination against MenB is recommended on an individual basis for adolescents and young adults [53,58].

Regarding Europe, on 14 January 2013, EMA licensed Bexsero-4CMenB. This vaccine provides protection against 73% to 87% of the circulating serogroup B strains. In September 2015, the UK became the first country to offer Bexsero to all infants free of charge.

Currently, Bexsero is recommended from >2 months of age in routine childhood national immunization programs, with different vaccination schedules in Austria, Italy, the Netherlands, the UK, the Republic of Ireland, and Lithuania. In these latter countries, unlike other European countries, vaccination is offered free of charge to all infants older than 2 months [53,55,56].

In Italy, the Italian Medicines Agency (AIFA) authorized the use of both Bexsero and Trumenba from 2014 (in infants >2 months) and 2017 (in children >10 years of age), respectively [59].

According to the “lifetime immunization schedule”, from 2014, the use of Bexsero in infants between 2 and 5 months of age was initially approved with a 3-dose vaccine immunization schedule at 3, 4, and 6 months, with each dose separated by at least 4 weeks, followed by 1 catch-up dose between 12 and 15 months of age [51,52].

However, some clinical trials showed that in infants younger than 12 months, a reduced 2-dose primary schedule is equally effective in terms of immunogenicity and safety compared to the classic schedule. Therefore, this schedule is currently recommended in Italy and has been offered as part of the publicly funded national routine immunization program since January 2017 free of charge for all infants from 2 months to 2 years of age [60].

The reduced 2-dose primary schedule plus a booster dose should indeed protect infants and toddlers for at least 2 years after the completion of the schedule. The adjusted vaccine effectiveness against all meningococcal group B strains for a single dose of Bexsero was 24.1%, which is consistent with the lack of an effect observed among infants who were 9–17 weeks of age. Vaccine effectiveness was 52.7% among children who received 2 priming doses of Bexsero while it was 71.2% among those who received 3 doses.

Three years after implementation, the protection of Bexsero against IMD caused by serogroup B among children was verified once again. For infants aged 6 to 11 months, a series of 2 primary doses followed by a booster in the second year of life (8 weeks between 1st and 2nd doses; 3rd dose at 12 months of age or 8 weeks after 2nd dose, whichever is later) is recommended. For toddlers aged 2 to 10 years, a 2-dose primary series is recommended as Bexsero has been proven to be immunogenic after at least 2 doses (8 weeks between doses). In adolescents and adults, 2 doses are needed (8 weeks between doses) [56,60,61,62].

Regarding Trumenba, the primary immunization program either adopts a 2-dose schedule (6 months between doses) or a 3-dose schedule (0, 1, and 6 or 0, 2, and 6 months), with a booster dose recommended for high-risk groups. The two-dose vaccination schedule is considered the gold standard. Furthermore, according to the “lifetime immunization schedule”, the MenB vaccine must be actively offered to high-risk subjects due to the presence of concomitant diseases (see Table 2), depending on work activity (e.g., operators who work in microbiology laboratories exposed to *Neisseria meningitidis*), and when epidemic outbreaks occur among the close contacts of affected subjects.

### 3.4. Men B Vaccine and “the Adolescent Problem”

According to epidemiological data on the incidence and prevalence of IMD and on the status of “nasopharyngeal carrier”, attention has been focused on the adolescent population, with the aim of convincing health authorities to implement a free MenB vaccination program for this age group. Adolescents have the highest rate of colonization, being the main reservoir and the major source of transmission of *N. meningitis* serogroup B. Indeed, asymptomatic meningococcal carriage is recognized as an age-dependent phenomenon.

The main risk factors for the acquisition of a new meningococcal serogroup and the development of a “healthy carriage status” (similar for any meningococcal serogroup) are active social habits, later year of schooling, smoking, respiratory tract infections, attending pubs or clubs, participation in intimate kissing, having had sexual intercourse (in particular same-sex intercourse), and illicit drug consumption. 

Another problem that primarily affects the adolescent population is the possible development of outbreaks. In the last 10 years, in the US, some outbreaks in college campuses that were primarily due to serogroup B have been reported [57,63].

Regarding Italy, according to a study carried out between January and March 2016 in Milan by Terranova et al., 5.3% of the population aged 14–21 years were carriers (2560 otherwise healthy students were enrolled) [64].

Whereas, as demonstrated in a study conducted in Genoa by Gasparini et al., which collected 200 samples from February to May 2011, 18.5% of adolescents were carriers and serogroup B was the most common [65].

In 2015–2016, there was an outbreak of IMD due to *N. meningitidis* serogroup C in Tuscany, Italy, after which a cross-sectional-survey was conducted to assess the meningococcal carriage prevalence, which was 4.8% overall, with a peak of 23.7% in 19 years old. Serogroup B was the most prevalent (1.8%) [66].

In Italy, from 2011 to 2017, the percentage of IMD due to *N. meningitidis* B among teens (10–14 years) and young people (15–24 years) was 28% and 32%, respectively; in 2018, it was 42.9% and 51.7% of the total reported cases, respectively [49].

Furthermore, the lethality of meningococcal disease tends to be higher among adolescents than children because the specific symptoms of meningococcal disease tend to appear later and have less rapid disease progression, causing adolescents to delay seeking medical care (22 h against 13–14 h in subjects 1–4 years) [67].

## 4. Discussion

The primary endpoints of a vaccine in general are first individual protection against the specific disease, and second disease prevention both in unimmunized people through a reduction in pharyngeal carriage and immunized people through a decrease in transmission (herd immunity, limiting the reservoir of the bacterium). Finally, a public health benefit in terms of cost-effectiveness through the definition of the best vaccination strategy is also achieved.

Concerning the meningococcal vaccines B, while the first aim has been effectively achieved through immunization programs, no appreciable effect of 4CMenB on the carriage prevalence has been obtained, resulting in a loss of herd protection. This is a problem, particularly among adolescents, who typically have the highest carriage rate. In this context, immunization through a vaccination program targeting adolescents may reduce the risk of late cases in some outbreaks, but it is unlikely to reduce transmission given the lack of an effect on carriage status [61,68]. Another possible aspiration is cross-protection with other meningococcal serogroups. Since the protein components included within both MenB vaccines can be expressed by other meningococcal serogroups, scientists thought that the development of vaccines directed against serogroup B could provide cross-protection against non-B serogroups while also reducing the carriage prevalence or the acquisition of strains of any capsular group. However, studies have not provided reassuring data on the antigens contained in the vaccine, which are only expressed by some of the B strains in circulation [69,70]. Vaccination programs should offer the MenB vaccine to infants under 1 year of age who are at the highest risk of infection and then should also offer the MenB vaccine free of charge to adolescents not previously vaccinated, co-administering it with others proposed in the same age group in the interest of improving compliance and adhesion.

About MenC vaccines, countries should improve the use of the MenACWY as the vaccine of choice instead of MenC in the first years of life. In this way, it could enhance protection against the W and Y strains, which are spreading more than in the past and protect against escape. Immunization against multiple meningococcal serogroups allows better coverage against IMD and provides an advantage in terms of costs. Vickers et al. demonstrated that the highest reduction in IMD incidence is obtained through the immunization of infants at 12 months of life with the quadrivalent vaccine, followed by a booster dose at 15 years of age. The administration of the monovalent vaccine at 12 months not only has a lower impact on IMD incidence but also causes the strain replacement to appear after 10 years of continuous use [45,71]. The switch from the monovalent to the quadrivalent vaccine in the vaccination schedule for infants could be the first aim. However, the promotion of the meningococcal ACYW vaccine for all subjects aged 11–12 years followed by a booster dose for adolescents aged 16 years is equally important to achieve lasting protection against all the serogroups.

An important problem in the development of a meningococcal vaccine is represented by the constant genetic mutations of the *N. meningitis* genome that determine antigenic and allelic variants and therefore the potential loss of long-term protection due to direct selective vaccination pressure. Indeed, meningococci can switch their polysaccharide capsules and switch off capsular expression, with an increased risk of vaccine evasion and antibiotic resistance (related to the bacteria’s propensity to acquire virulence factors) [25].

## 5. Looking to the Future

One of the most effective strategies for implementing vaccination programs and achieving adequate vaccination coverage is a specific Health Technology Assessment (HTA) report, recognized as the most excellent approach to evaluating the introduction of new vaccination strategies in prevention programs. According to the WHO and the “lifetime immunization schedule” 2017–2019, HTA is the best channel that pursues these criteria. Indeed, HTA periodically carries out a systematic review of the scientific evidence to correctly outline the impact of the introduction of new vaccines and new vaccination strategies.

The cost-effectiveness of a vaccine strategy is mainly associated with the incidence and high costs of the natural disease (especially the long-term ones), the vaccine’s effectiveness and its price, and the duration of the protection conferred through the immunization schedule [68,72].

First, to improve coverage, it is necessary to identify an effective communication strategy to promote adhesion and active participation in the vaccination campaign. Since, in this context, the target population is adolescents, excellent instruments for spreading information include the web, social media, and social networks, which are increasingly usable and widespread. On the other hand, it is essential to warn the population about the risk of unreliable and incorrect information [68].

Furthermore, an element in support of the meningococcal vaccination campaign is certainly represented by the fact that the related disease is known and perceived to be serious, with high lethality and important sequelae. This perception could significantly increase the compliance with and response to the vaccination campaign; consequently, proper education about the severity of IMD, its sequelae, and methods of acquisition and transmission is essential [73,74,75,76].

As demonstrated by several studies in the literature, to optimize the organizational process and adherence “in the adolescent context”, a valid immunization platform is the administration of meningococcal vaccines in schools [77,78,79,80].

In a cost-effective-based vaccination strategy, to enhance compliance, another excellent aspect is the chance of co-administering the MenB vaccine with other vaccines proposed in adolescence, such as the meningococcal ACYW vaccine, Tdap-IPV and Tdap vaccines, or HPV vaccine. Since the nonavalent anti-HPV vaccine has recently been approved, data about its co-administration with the MenB vaccine are not yet available [81]. 

Finally, a universal meningococcal vaccine, especially one that incorporates the capsular polysaccharide and surface protein antigens, is necessary; an economical pentavalent meningococcal conjugate vaccine (MenACWYX, NmCV-5, Serum Institute of India) is in development for use in Africa. This vaccine includes serogroup A and X polysaccharides one by one conjugated to TT and serogroup C, W, and Y polysaccharides individually conjugated to CRM. Tapia et al. suggested that the NmCV-5 pentavalent vaccine could provide immunity in toddlers after a single dose, without safety concerns. An investigational meningococcal ABCWY vaccine (MenABCWY) that is made up of components of vaccines against meningococcal serogroup B (4CMenB) and serogroups ACWY (MenACWY) is being developed; recent studies demonstrated its immunogenicity and its safety profile in adolescents [82,83].

While waiting for a universal meningococcal vaccine that provides coverage for all serogroups, the vaccine recommendations could be aligned among countries to obtain homogenous vaccine schedules. 

## Figures and Tables

**Table 1 ijerph-19-04035-t001:** Currently licensed and available against serogroups A, C, W, and Y vaccines.

Vaccine	Protein-Carrier	Year of License	Licensed to Use	Schedule	Simultaneous Co-Administration with Other Vaccines
MenACWY-D, MenActra Sanofi Pasteur SA, Lyon, France	Diphtheria toxoid	2005	9 months–55 years of age	PRIMARY VACCINATION: -Infants 9–23 months of age: 2 doses (3 months between 1st and 2nd doses) -Individuals 2–55 years of age: A single dose. BOOSTER DOSE: -A single booster dose may be administered to people 15–55 years of age at risk of meningococcal disease (4 years after the conclusion of the primary vaccination schedule).	MMRV, MMR + V, PCV7, HAV vaccine, Typhum Vi, Td, DTaP + IPV
MenACWY-CRM, Menveo GlaxoSmithKline Biologicals SA, Rixensart Belgium	Diphtheria protein cross-reactive material 197	2010	2 months-55 years of age	PRIMARY VACCINATION: -Infants aged 2 months: 4-dose series at 2, 4, 6, and 12 months of age. -Children aged 7–23 months: Two doses, with the second dose administered in the second year of life and at least 3 months after the first dose. -Children aged 2–10 years: A single dose. A single booster dose may be administered 2 months after the first dose to children aged 2–5 years at risk of meningococcal disease. -Adolescents and adults 11–55 years of age: A single dose. BOOSTER DOSE: -Adolescents and adults 15–55 years of age: A single booster dose may be administered to people 15–55 years of age at risk of meningococcal disease (4 years after the conclusion of the primary vaccination schedule with serogroups A, C, Y, and W-135 conjugate vaccine).	Tdap, HPV, MMR, MMR + V, PCV7, pentavalent rotavirus, DTaP-HBV-IPV/Hib vaccine, HAV vaccine
MenACWY-TT, NimenrixPfizer Inc., New York, NY, USA	Tetanus toxoid	2012	6 weeks–55 years of age	PRIMARY VACCINATION: -Infants 6 weeks–6 months of age: 2 doses (2 months between the 1st and 2nd doses) -Infants > 6 months of age, children, adolescents, and adults: A single dose. An additional primary dose of Nimenrix may be indicated in some specific cases. BOOSTER DOSE: -Infants 6 weeks–12 months of age: a booster dose should be given at 12 months of age, at least 2 months after the last dose of primary immunization schedule. In previously vaccinated children >12 months of age, a booster dose of Nimenrix could be administered if they have received primary vaccination with a conjugated or plain polysaccharide meningococcal vaccine.	DTaP-HBV-IPV/Hib vaccine, PCV10, hepatitis A (HAV), hepatitis B (HBV) vaccines, MMR, MMR + V, unadjuvanted seasonal influenza vaccine, DTaP, DTaP-HBVIPV/Hib, PCV-13, HPV2
MenACWY-TT, MenQuadfi Sanofi Pasteur SA, Lyon, France	Tetanus toxoid	2020	>2 years of age	PRIMARY VACCINATION: -Subjects >2 years of age: A single dose. BOOSTER DOSE: A single dose of MenQuadfi may be administered to individuals >15 years of age at risk of meningococcal disease (4 years after the conclusion of the primary vaccination schedule with serogroups A, C, Y, and W-135 conjugate vaccine).	Tdap and HPV

Polysaccharide diphtheria toxoid conjugate vaccine (MenACWY-D); oligosaccharide diphtheria protein cross-reactive material 197 conjugate vaccine (MenACWY-CRM); polysaccharide tetanus toxoid conjugate vaccine (MenACWY-TT); Diphtheria, tetanus, and pertussis (DTaP); *Haemophilus influenzae* type b (Hib); 7-valent pneumococcal conjugate (PCV7); 10-valent pneumococcal conjugate (PCV10); 13-valent pneumococcal conjugate (PCV13); inactivated poliovirus vaccine (IPV), hepatitis B virus (HBV); hepatitis A virus (HAV); adult diphtheria, tetanus and pertussis (Tdap); Human papillomavirus vaccine (HPV); measles, mumps, rubella, and varicella virus vaccine (MMRV); measles, mumps, and rubella virus vaccine (MMR); varicella virus vaccine (V).

**Table 2 ijerph-19-04035-t002:** People at increased risk of meningococcal disease.

People at risk because of a serogroup A, B, C, W, or Y meningococcal disease outbreak
HIV infection, congenital immunodeficiencies, and type 4 toll-like receptor defects and defects in properdin
Anatomical or functional asplenic people, including people with sickle cell disease and thalassemia
Clinical conditions characterized by immunosuppression status (such as organ transplantation o antineoplastic therapy, including systemic corticosteroid therapy in high doses)
Congenital complement defects (C5–C9) and use of complement inhibitor drugs, such as eculizumab or ravulizumab
Chronic diseases (diabetes mellitus type 1, renal insufficiency with creatinine clearance <30 mL/min, severe chronic liver disease)
Loss of cerebrospinal fluid
Microbiologists who routinely work with isolates of N. meningitidis
People traveling to or living in areas of the world where N. meningitidis is endemic (such as some regions in Africa)
College students who live in residence halls and have not been fully vaccinated with the meningococcal ACWY vaccine
US military recruits

## Data Availability

Not applicable.

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
