# Peer review of "Meningococcal Disease in Pediatric Age: A Focus on Epidemiology and Prevention"

_ijerph, 2022, doi:10.3390/ijerph19074035_

Round 1

Reviewer 1 Report

This manuscript is a review of meningococcal epidemiology and vaccine prevention. It is an important topic given the focus of WHO’s Defeating Meningitis by 2030 global road map and the content is appropriate though it could definitely be improved in a number of areas:

  1. Organizationally, it might be better to have the major titles as introduction, epidemiology, vaccines, conclusions and looking to the future and then have sub-titles within the vaccine section on introduction (which would discuss approaches – PS, conjugates, proteins), then monovalent conjugate vaccines, quadrivalent conjugate vaccines, MenB vaccines and then maybe new vaccines (ACYW+B and ACYW+X).
  2. It is not clear the difference between Table 2 and Table 3 since they have the same title. Could they be combined into a single table?
  3. Table 4 doesn’t really warrant being a table. The three points can be just bulleted/numbered in the text.
  4. The focus of the manuscript when discussing vaccine policy and impact seems to be primarily US and Europe with some mention of Africa but virtually no mention of Latin America or Asia. Either add a section on these regions or explain the focus of this article.
  5. In the section on quadrivalent meningococcal conjugate vaccines, it seems a little disjointed. It might be best to separate out by geographies i.e. UK, Europe, US as it seems to go back and forth and sometimes it is hard to tell what the geography is being discussed without having to check the reference (e.g. lines 226-230).
  6. I suggest to remove “conjugate” from the title when discussing serogroup B vaccines as it may be confusing.
  7. The section entitled “Conclusions” seems more like a discussion.
  8. As discussed by the authors, there are many different vaccine schedules in different parts of the world, some of which is based on local epidemiology. It would be good discuss ways of harmonization for global recommendations.

Author Response

The authors thank the reviewers for his/her valuable suggestions. The manuscript has been amended accordingly.

  1. Organizationally, it might be better to have the major titles as introduction, epidemiology, vaccines, conclusions and looking to the future and then have sub-titles within the vaccine section on introduction (which would discuss approaches – PS, conjugates, proteins), then monovalent conjugate vaccines, quadrivalent conjugate vaccines, MenB vaccines and then maybe new vaccines (ACYW+B and ACYW+X). We divided the article, as suggested, into five major titles, respectively: introduction, epidemiology, vaccines, discussion and looking to the future. We added sub-titles within the vaccine section (introduction, monovalent meningococcal conjugate vaccines, quadrivalent meningococcal conjugate vaccines, vaccines against serogroup B and finally Men B vaccine and “the adolescent problem”). We preferred to leave the mention of new vaccines (ACYW+B and ACYW+X) in the paragraph “looking to the future”, because little data are currently available about their development.
  2. It is not clear the difference between Table 2 and Table 3 since they have the same title. Could they be combined into a single table? The authors combined the two tables into a single table, as recommended.
  3. Table 4 doesn’t really warrant being a table. The three points can be just bulleted/numbered in the text. The authors incorporated the three points into the text and removed the table 4.
  4. The focus of the manuscript when discussing vaccine policy and impact seems to be primarily US and Europe with some mention of Africa but virtually no mention of Latin America or Asia. Either add a section on these regions or explain the focus of this article. The authors added in the introduction the focus of the article, that was centered on developed countries (Europe and North America) and less on low-income countries, especially Africa. The authors preferred not to be extremely dispersive, so decided to focus on the US, Europe (the UK and Italy), with some mentions about Africa.
  5. In the section on quadrivalent meningococcal conjugate vaccines, it seems a little disjointed. It might be best to separate out by geographies i.e. UK, Europe, US as it seems to go back and forth and sometimes it is hard to tell what the geography is being discussed without having to check the reference (e.g. lines 226-230). The authors followed the suggestion and separated the section into three sub-paragraph: US, Europe (the UK) and Italy. The authors reported for eah country both vaccination schedules and the consequently reduction in the incidence of the disease after vaccine’s introduction.
  6. I suggest to remove “conjugate” from the title when discussing serogroup B vaccines as it may be confusing. Done.
  7. The section entitled “Conclusions” seems more like a discussion. The authors substituted it.
  8. As discussed by the authors, there are many different vaccine schedules in different parts of the world, some of which is based on local epidemiology. It would be good discuss ways of harmonization for global recommendations. The authors added a sentence about it in the last paragraph of the article.

Finally, the manuscript underwent English revision.

Reviewer 2 Report

This is a well-written comprehensive review that focuses  on the epidemiology of meningococcal disease and its prevenion through vaccination in children and adolescents.

 In the abstract section, I would like to see the aim of the review. The abstract looked more like an introduction. 

In line 350, it is mentioned that Bextero is offered free of charge to all infants older than two months in Greece. However, this is not true as Bextero is not included in the Greek National Immunization Schedule and is definitely not offered free of charge.

The caption of table 2 should be changed to "People at increased risk for meningococcal disease by serogroups A, C, W, Y".

he caption of table 3 should be changed to "People at increased risk for meningococcal disease by serogroup B".

I suggest that the two tables (2 and three) are rather replaced by one table that would contain all known risk factors for meningococcal disease.

Author Response

The authors thank the reviewers for his/her valuable suggestions. The manuscript has been amended accordingly.

  1. In the abstract section, I would like to see the aim of the review. The abstract looked more like an introduction. As suggest, the authors not only added the aim of the review in the abstract (lines 19-26), but also added the same purposes in the “discussion” paragraph.
  2. In line 350, it is mentioned that Bextero is offered free of charge to all infants older than two months in Greece. However, this is not true as Bextero is not included in the Greek National Immunization Schedule and is definitely not offered free of charge. The authors thank the reviewer for the clarification and confirmed the correction of the manuscript.
  3. The caption of table 2 should be changed to "People at increased risk for meningococcal disease by serogroups A, C, W, Y". The caption of table 3 should be changed to "People at increased risk for meningococcal disease by serogroup B". I suggest that the two tables (2 and three) are rather replaced by one table that would contain all known risk factors for meningococcal disease. The authors combined the two tables into a single table, as recommended.

Finally, the manuscript underwent English revision.